# Cooperation among *c*-subunits of F$_o$F$_1$-ATP synthase in rotation-coupled proton translocation

Noriyo Mitome[1,2,3]*†, Shintaroh Kubo[4]†, Sumie Ohta[2], Hikaru Takashima[3], Yuto Shigefuji[3], Toru Niina[4], Shoji Takada[4]*

[1]Faculty of Education, Tokoha University, Shizuoka, Japan; [2]Department of Chemistry and Biochemistry, National Institute of Technology, Numazu College, Numazu, Japan; [3]Department of Chemical and Biological Engineering, National Institute of Technology, Ube College, Ube, Japan; [4]Department of Biophysics, Graduate School of Science, Kyoto University, Kyoto, Japan

**Abstract** In F$_o$F$_1$-ATP synthase, proton translocation through F$_o$ drives rotation of the *c*-subunit oligomeric ring relative to the *a*-subunit. Recent studies suggest that in each step of the rotation, key glutamic acid residues in different *c*-subunits contribute to proton release to and proton uptake from the *a*-subunit. However, no studies have demonstrated cooperativity among *c*-subunits toward F$_o$F$_1$-ATP synthase activity. Here, we addressed this using *Bacillus* PS3 ATP synthase harboring a *c*-ring with various combinations of wild-type and *c*E56D, enabled by genetically fused single-chain *c*-ring. ATP synthesis and proton pump activities were decreased by a single *c*E56D mutation and further decreased by double *c*E56D mutations. Moreover, activity further decreased as the two mutation sites were separated, indicating cooperation among *c*-subunits. Similar results were obtained for proton transfer-coupled molecular simulations. The simulations revealed that prolonged proton uptake in mutated *c*-subunits is shared between two *c*-subunits, explaining the cooperation observed in biochemical assays.

*For correspondence:
mitome@sz.tokoha-u.ac.jp (NM);
takada@biophys.kyoto-u.ac.jp
(ST)

†These authors contributed
equally to this work

Competing interest: The authors
declare that no competing
interests exist.

Reviewing Editor: Nir Ben-Tal,
Tel Aviv University, Israel

## Editor's evaluation

F$_o$F$_1$-ATP synthase is a membrane enzyme that uses the proton motive force to synthesize ATP via rotation-coupled mechanism. The symmetry of this high-order homo-oligomeric enzyme suggests cooperativity among the protomers. Here, biochemical and molecular biology studies, and simulations demonstrate that energy transduction is indeed inherently cooperative in F$_o$F$_1$-ATP synthase.

## Introduction

F$_o$F$_1$-ATP synthase (F$_o$F$_1$) is a ubiquitous enzyme that synthesizes or hydrolyzes ATP coupled with proton translocation at the inner mitochondrial membrane, chloroplast thylakoid membrane, and bacterial plasma membrane (*Boyer, 1997*; *Walker, 2013*; *Yoshida et al., 2001*). F$_o$F$_1$ synthesizes ATP via rotation of the central rotor driven by the proton motive force across the membrane. The enzyme comprises two rotary motors that share the rotor, that is, the water soluble F$_1$, which has catalytic sites for ATP synthesis/hydrolysis (*Noji et al., 2017*), and the membrane-embedded F$_o$, which mediates proton translocation (*Kühlbrandt, 2019*). The F$_o$ motor consists of a *c* oligomer ring (*c*-ring), which serves as the rotor, and the *ab$_2$* stator portion located on the *c*-ring periphery. Downgradient proton translocation through F$_o$ drives rotation of the central rotor composed of a *c*-ring and γε subunits,

**eLife digest** Cells need to be able to store and transfer energy to fuel their various activities. To do this, they produce a small molecule called ATP to carry the energy, which is then released when the ATP is broken down. An enzyme found in plants, animals and bacteria, called $F_oF_1$ ATP synthase, can both create and use ATP. When it does this, protons, or positive hydrogen ions, are transported across cellular boundaries called membranes.

The region of the enzyme that is responsible for pumping the protons contains different parts known as the c-ring and the a-subunit. The movement of protons drives the c-ring to rotate relative to the a-subunit, which leads to producing ATP. Previous research using simulations and the protein structures found there are two or three neighbouring amino acids in the c-ring that face the a-subunit, suggesting that these amino acids act together to drive the rotation.

To test this hypothesis, Mitome et al. mutated these amino acids to examine the effect on the enzyme's ability to produce ATP. A single mutation reduced the production of ATP, which decreased even further with mutations in two of the amino acids. The extent of this decrease depended on the distance between the two mutations in the c-ring. Simulations of these changes also found similar results. This indicates there is coordination between different parts of the c-ring to increase the rate of ATP production.

This study offers new insights into the molecular processes controlling ATP synthesis and confirms previous theoretical research. This will interest specialists in bioenergetics because it addresses a fundamental biological question with broad impact.

thereby inducing conformational changes in $F_1$ that result in ATP synthesis. Conversely, ATP hydrolysis in $F_1$ induces reverse rotation of the rotor, which forces $F_o$ to pump protons in the reverse direction.

The c-ring is composed of 8–17 c-subunits depending on the species (*Watt et al., 2010*; *Stock et al., 1999*; *Mitome et al., 2004*; *Meier et al., 2005*; *Matthies et al., 2009*; *Vollmar et al., 2009*; *Pogoryelov et al., 2009*). $F_oF_1$ from thermophilic *Bacillus* PS3 and yeast mitochondrial $F_oF_1$ contain 10 c-subunits in the c-ring, which is designated the $c_{10}$-ring (*Stock et al., 1999*; *Mitome et al., 2004*; *Symersky et al., 2012*; *Guo et al., 2019*; *Figure 1*). The $F_o$-c subunit harbors an essential proton-binding carboxyl group (c-Glu; cE56 in *Bacillus* PS3, cE59 in yeast mitochondria) located near the center of the membrane-embedded region; this group functions as the proton carrier (*Figure 1a*). Protonation in the Glu allows the $c_{10}$-ring to bind a proton, whereas proton release leads to Glu deprotonation. Accordingly, bacterial $F_oF_1$ activity is significantly decreased when the corresponding key residue is modified by the inhibitor *N,N*-dicyclohexylcarbodiimide (DCCD) (*Hermolin and Fillingame, 1989*) or mutated to other amino acids (*Dmitriev et al., 1995*), and *Bacillus* PS3 $F_oF_1$ carrying a single cE56Q mutation in the $c_{10}$-ring does not catalyze ATP-driven proton pumping or ATP synthesis (*Mitome et al., 2004*).

The a-subunit comprises two separate half-channels, one connecting the c-ring to the periplasm side of the bacteria or the intermembrane space side of the mitochondria, the other connecting the c-ring to the cytoplasmic side of the bacteria or the matrix side of the mitochondria (*Figure 1b*). Recent cryo-electron microscopy (EM) structural analyses of $F_oF_1$ at near-atomic resolution (*Allegretti et al., 2015*; *Zhou et al., 2015*) have revealed two long tilted parallel α-helices in the a-subunit at the interface with the $c_{10}$-ring. An essential Arg residue (aR169 in *Bacillus* PS3, aR176 in yeast mitochondria) at the middle of the long parallel helices plays a critical role in separating the two half-channels by preventing proton leakage (*Mitome et al., 2010*), and in the half-channels, two highly conserved Glu residues (aE223 and aE162 in yeast mitochondria) are regarded as proton-relaying sites (*Srivastava et al., 2018*; *Figure 1a*). Since the essential Arg (a-Arg) localizes near c-Glu in the $c_{10}$-ring, the attractive interaction between a-Arg and deprotonated c-Glu is hypothesized to also contribute to $F_o$ rotation (*Vik and Antonio, 1994*).

In the $F_o$ rotation models proposed based on experimental studies (*Vik and Antonio, 1994*; *Elston et al., 1998*; *Kubo et al., 2020*), the c-subunits facing the a-subunit perform three functions (proton release, electrostatic interaction with a-Arg, and proton uptake) depending on their positions relative to the a-subunit. A high-resolution structure analysis of yeast mitochondrial $F_oF_1$ showed four of the 10 c-subunits to be facing the a-subunit (*Srivastava et al., 2018*). Three key residues, that is, aGlu162,

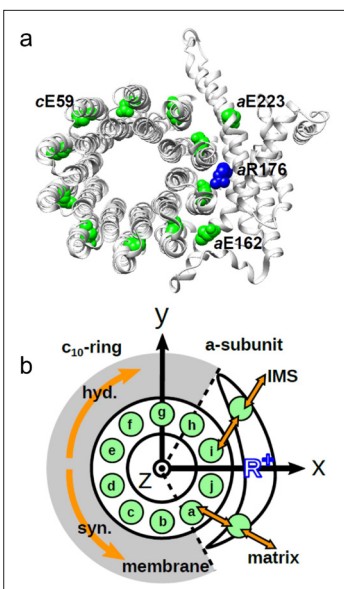

**Figure 1.** Schematic picture of the *a*-subunit and *c*-ring of $F_o$. (**a**) The $ac_{10}$ part of the $F_o$ region is depicted as a ribbon diagram. Spheres represent *c*E59, which was substituted in this study, *a*E223, *a*E162, and *a*R176 (blue) (the residue numbers are those from yeast). (**b**) Schematic diagram of our simulation model. Green circles represent protonatable glutamates. Those in the 10 *c*-subunits are labeled (a–j). The membrane drawn in gray is modeled implicitly. Protons can hop between *c*E59 and the glutamates in the *a*-subunit, *a*E223, and *a*E162. Additionally, *a*E223 and *a*E162 exchange their protons with the inner membrane space (IMS) and matrix aqueous environment, respectively. Arrows in orange indicate the net proton flow. We set the rotational axis of the $c_{10}$-ring as the z-axis, and the position of *a*R176 as the x-axis. Clockwise rotation of the *c*-ring occurs in the ATP hydrolysis mode, and counterclockwise rotation of the *c*-ring occurs in the ATP synthesis mode.

The online version of this article includes the following source data for figure 1:

**Source data 1.** Schematic picture of the *a*-subunit and *c*-ring of $F_o$.

**Source data 2.** Schematic diagram of simulation model.

*a*R173, and *a*Glu223, localize between the *c*-Glu residues of the four *c*-subunits, suggesting that the *c*-Glu residues of adjacent *c*-subunits could cooperate through the *a*-subunit residues. A more recent theoretical study using a hybrid Monte Carlo/molecular dynamics (MC/MD) simulation based on a high-resolution structure showed that there can be two or three deprotonated *c*-Glu residues facing the *a*-subunit concurrently (*Kubo et al., 2020*). This suggests that the waiting time for protonation of *c*-subunits is shared among two or three *c*-subunits. However, the relationship between a shared deprotonation time among multiple *c*-subunits and their cooperation in proton transport remain to be characterized.

To directly investigate the cooperation among the *c*-subunits in the $c_{10}$-ring, we used a genetically fused single-chain *c*-ring and analyzed the function of *Bacillus* PS3 $F_oF_1$ carrying hetero *c*E56D mutations. Biochemical assays showed that the ATP synthesis activity was reduced, but not completely inhibited, by a single *c*E56D mutation, and that it was further reduced by double *c*E56D mutations. Importantly, across all five double mutants, the activity tended to decrease further as the distance between the two mutation sites increased. To clarify the underlying molecular mechanisms, we performed proton transfer-coupled MD simulations of $F_o$, in which the mutations were mimicked, reproducing the characteristics of the biochemical experiment. From the analysis of the simulation trajectories, we found that prolonged duration times for proton uptake in the two mutated *c*-subunits can be shared. As the distance between the two mutation sites increases, the degree of time-sharing decreases. Taken together, these results reveal the functional coupling between neighboring *c*-subunits.

## Results

### Biochemical assays using $F_oF_1$s with a fused *c*-ring harboring hetero mutations

To investigate potential cooperation among the *c*-subunits in the $c_{10}$-ring rotation driven by proton translocation, we generated $F_oF_1$ mutants harboring a hetero-mutated $c_{10}$-ring from thermophilic *Bacillus* PS3. We previously produced a fusion mutant, $c_{10}$ $F_oF_1$, in which 10 copies of the $F_o$-*c* subunit in the $c_{10}$-ring were fused into a single polypeptide, and demonstrated that $c_{10}$ $F_oF_1$ was active in proton-coupled ATP hydrolysis/synthesis (*Mitome et al., 2004*). Starting with $c_{10}$ $F_oF_1$, we generated six mutant $F_oF_1$s harboring one or two hetero *c*E56D-mutated *c*-subunits. The single mutant carries a *c*E56D mutation in the *c*(e)-subunit (dtesignated as mutant "e"), whereas the five double mutants, "ef," "eg," "eh," "ei," and "ej," harbor two *c*E56D mutations, each with its respective *c*-subunit (*Figure 1b*).

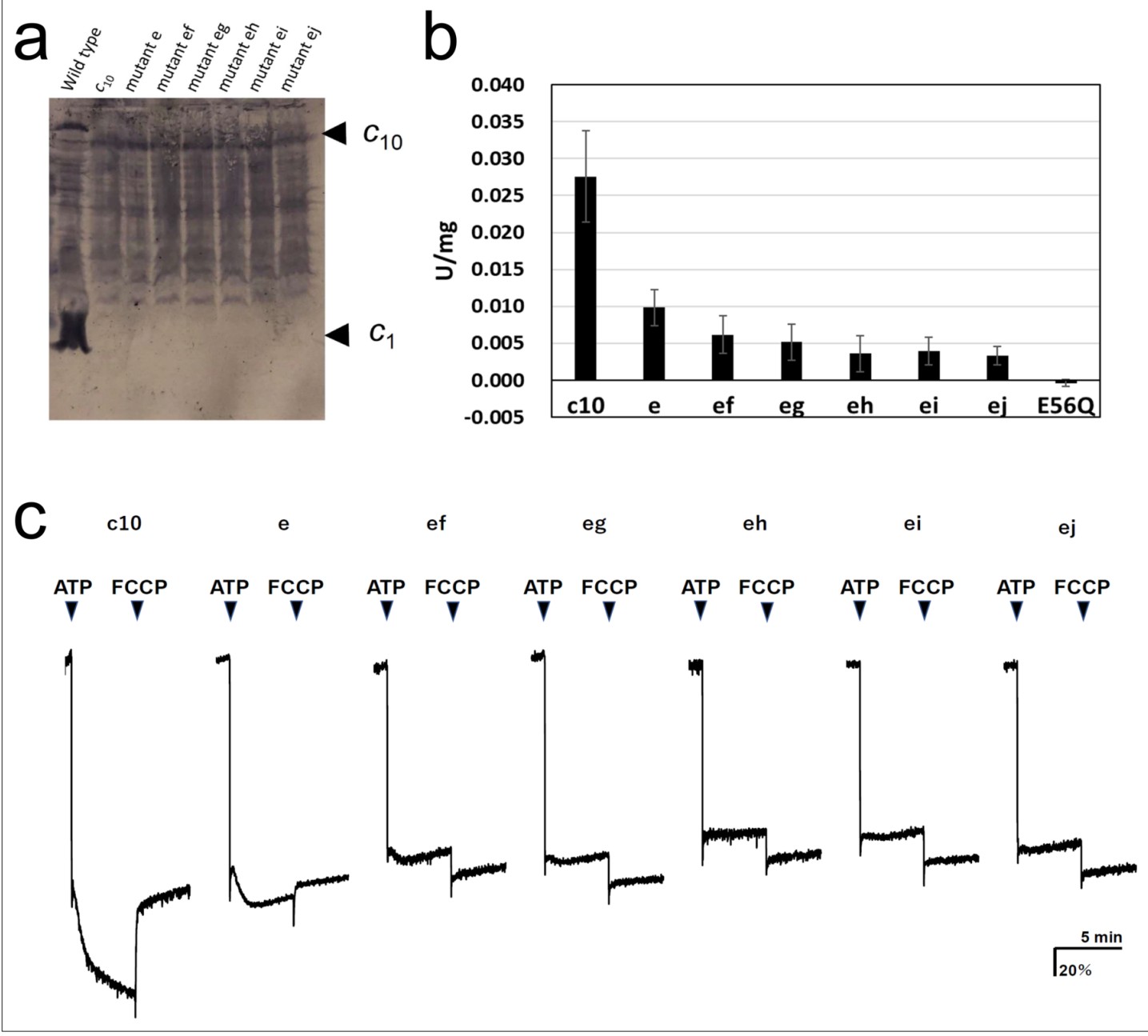

**Figure 2.** Expression of the mutated $F_o$-$c$ subunit and proton pump, and ATP synthesis activities of membrane vesicles containing mutated $F_oF_1$s. (**a**) Proteins were separated using SDS-PAGE and immunoblotted with anti-$F_o$-$c$ antibodies. (**b**) ATP synthesis driven by NADH oxidation. The rightmost bars [E56Q] show the results of $c_{10}$(E56Q)-$F_oF_1$. Error bar: standard error. (**c**) ATP-driven proton pump activity was measured by monitoring ACMA fluorescence quenching.

The online version of this article includes the following source data for figure 2:

**Source data 1.** Expression of the mutated $F_o$-*c* subunit.

**Source data 2.** ATP synthesis activities of membrane vesicles containing mutated $F_oF_1$s.

$F_oF_1$ mutants carrying one or two *c*E56D substitutions in the $c_{10}$-ring were expressed in host *Escherichia coli* cell membranes at approximately one-tenth the level of wild-type (WT) $F_oF_1$. Western blotting with anti-*c*-subunit antibodies showed $c_{10}$-subunit expression in all mutants (**Figure 2a**). Unlike in the WT, there was no band of monomer *c*-subunits in the mutants, and relatively stronger bands were seen at the position of the $c_{10}$ subunit, indicating that the $c_{10}$ subunit of the mutants was expressed

**Table 1.** P-value of ATP synthesis activity between the two mutants.

|       | e | ef | eg | eh | ei | ej |
|-------|---|----|----|----|----|----|
| $c_{10}$ | $5.41\times10^{-7}$ | $1.08\times10^{-8}$ | $6.44\times10^{-9}$ | $2.45\times10^{-9}$ | $1.88\times10^{-9}$ | $1.73\times10^{-8}$ |
| e     |   | $5.29\times10^{-3}$ | $9.14\times10^{-4}$ | $3.54\times10^{-5}$ | $1.61\times10^{-5}$ | $6.25\times10^{-6}$ |
| ef    |   |   | 0.409 | 0.0357 | 0.0435 | 0.0122 |
| eg    |   |   |   | 0.179 | 0.241 | 0.0752 |
| eh    |   |   |   |   | 0.706 | 0.784 |
| ei    |   |   |   |   |   | 0.420 |

in the membrane (*Figure 2a*). First, ATP synthesis activity was measured using inverted membrane vesicles containing mutated $F_oF_1$s (*Figure 2b*).

The activity of mutant "e" was reduced to 35.6±8.8% of that of $F_oF_1$, with fusion mutation only. The activity of the five double mutants with fusion mutation only was lower than that of the single mutation ("ef": 22.3±9.3%; "eg": 18.8±8.8; "eh": 13.0±8.9%; "ei": 14.4±6.7%; "ej": 12.0±4.7%). The ATP synthesis activity of mutant "ef" was significantly higher than that of mutants "eh," "ei," and "ej"; the corresponding p-values were 0.0357, 0.0435, and 0.0122, respectively (*Table 1*). Since the ATP synthesis activity tended to decrease further as the distance between two mutation sites increased, a regression analysis among double mutations was performed between the distance between mutations, indicated by the number of *c*-subunits (ef=1, eg=2, eh=3, ei=4, and ej=5), and the ATP synthesis activity of these mutants. The regression confirmed that the ATP synthesis activity significantly decreased as the distance between the two mutations increased (p=0.0039). For comparison, the $c_{10}$(E56Q)-$F_oF_1$ with only a single E56Q mutation introduced into the first hairpin unit of the $c_{10}$ did not catalyze ATP synthesis (*Mitome et al., 2004*). Next, ATP-driven proton pump activity was assessed as a measure of the quenching of the fluorescence of 9-amino-6-chloro-2-methoxyacridine (ACMA) caused by proton influx into the inverted membranes (*Figure 2c*).

Mutants "e," "ef," and "eg" showed proton pumping, indicated by the slow quenching after ATP addition, while mutants "eh," "ei," and "ej" did not show any pumping. The proton pump activity of the single mutant "e" was higher than that of the double mutants "ef" and "eg." Thus, proton pump activity was high in the double mutants "ef" and "eg," in which the two mutations were located close to each other, but low in "eh," "ei," and "ej," in which the mutations were introduced farther apart. Although the mutant $F_oF_1$s showed ATP hydrolysis activity, approximately 90% of the activity was insensitive to DCCD, a compound that inhibits $F_o$ (*Table 2*). DCCD-insensitive ATP hydrolysis indicates uncoupled $F_oF_1$ activity. All mutants showed 10–15% DCCD-sensitive ATP hydrolysis activity. Thus, a subtle difference in the structure of the proton-binding site induced by the *c*E56D mutation may have conferred resistance to DCCD binding or caused uncoupling. The rotation driven by ATP hydrolysis was affected to a greater extent by the threefold symmetry structure of $F_1$ than by the rotation during synthesis, and the DCCD-sensitive ATP hydrolysis activity indirectly reflected the function of $F_o$.

**Table 2.** Membrane ATPase activity from cells expressing hetero-mutated *c*-subunits.

| Mutant | ATPase activity* | |
|--------|-------|-------|
|        | –DCCD | +DCCD |
| WT     | 0.23  | 0.076 |
| $c_{10\text{-fusion}}$ | 0.15 | 0.067 |
| Mutant e | 0.087 | 0.076 |
| Mutant ef | 0.090 | 0.065 |
| Mutant eg | 0.078 | 0.070 |
| Mutant eh | 0.086 | 0.073 |
| Mutant ei | 0.088 | 0.080 |
| Mutant ej | 0.083 | 0.068 |

*Membrane ATPase activity was measured after pre-incubation of membranes at 10 mg/ml in PA3 buffer with or without 50 µM DCCD for 20 min at 25°C. Activity is expressed as µmol/min/mg.

## MD simulation of hetero-mutated $F_oF_1$s

Biochemical assays showed that the decreased rotation speed of the double-mutant $F_o$ motor depended on the distance between the two mutation sites; however, the underlying mechanism remains to be elucidated. To obtain mechanistic insights, we tested the mutated $F_o$ motor

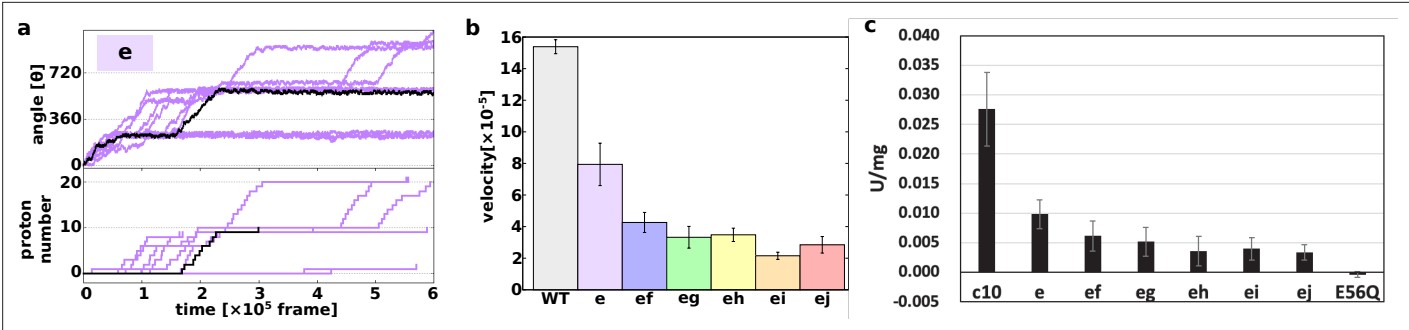

**Figure 3.** Proton transfer-coupled MD simulation of the WT and hetero mutants with Asp substitution of Glu. (**a**) Ten trajectories of the "e" mutant. The black line shows one representative trajectory. Upper part: rotation angle from initial position of $c$(a); lower part: the number of protons that entered from the IMS channel and were transported to the matrix channel through rotation. (**b**) Average rotational velocities for WT and mutants. Error bar: standard error. (**c**) ATP synthesis driven by NADH oxidation. In order to evaluate the correspondence between simulation data and experimental data, ATP synthesis activity which is identical to **Figure 2B** is shown. IMS, inner membrane space; MD, molecular dynamics; WT, wild-type.

The online version of this article includes the following source data and figure supplement(s) for figure 3:

**Source data 1.** Proton transfer-coupled MD simulation of the WT and hetero mutants with Asp substitution of Glu.

**Figure supplement 1.** (**a**) Simulation overview; (**b**) visualized $c$E59 (left) and $c$E59D (right); (**c**) the transfer efficiency under the effect of mutation.

**Figure supplement 2.** The average rotation velocities of WT and mutant $c$-rings in MD simulations with different parameter sets in the $c$E59D mutant.

rotations by proton transfer-coupled molecular simulations (**Kubo et al., 2020**). Based on our previous simulation setup for the WT yeast mitochondrial $F_o$, we introduced the $c$E59D mutation in silico to one and two $c$-subunits corresponding to the biochemical assays (see Materials and methods for more details).

First, we demonstrated 10 trajectories for the single mutant "e" (**Figure 3a**). Although the mutated $c_{10}$-ring paused for a long period, the mutants still rotated in the synthesis direction, coupled with proton transportation.

Next, we simulated all five double mutants ("ef," "eg," "eh," "ei," and "ej") and calculated the average rotational velocities over 10 trajectories (**Figure 3b**). **Figure 3b** shows the mean values and standard errors of the rotational velocities of the WT and all mutants. The rotational velocity of mutant "e" is almost two times slower than that of the WT. The rotational velocities of double mutants tend to decrease as the distance between the mutated chains increases. Thus, we were able to capture the characteristics of the experimental results in our simulations qualitatively, but not quantitatively.

We then evaluated the molecular processes for the simulation. Each $c$E59 (or $c$E59D) is protonated when the corresponding $c$-subunit is far from the $a$-subunit. This is regarded as the resting state of $c$E59 (**Figure 4a**). As counterclockwise rotation occurs, the $c$-subunit approaches the half-channel of the $a$-subunit, which is connected to the matrix (the matrix half-channel). When $c$E59 comes close to $a$E162, which is the relaying site to the matrix half-channel, proton transfer from $c$E59 to $a$E162 occurs via the Monte Carlo step. Depending on the transfer efficiency, several Monte Carlo steps may be required to achieve proton release from $c$E59. We define the time from the first trial of the $c$E59-to-$a$E162 proton transfer to the success of transfer as "the duration for proton release" (indicated in pink in **Figure 4a**). Once $c$E59 is deprotonated, the corresponding $c$-subunit can rotate counterclockwise further into the $a$-subunit facing region. After some rotation, the $c$-subunit approaches the other half-channel connected to the IMS (the IMS half-channel). When $c$E59 comes close to $a$E223, which is the relaying site for the IMS channel, $c$E59 attempts to take up a new proton from $a$E223 via the Monte Carlo step. We define the time from the success of proton release to the arrival at the rotation angle for proton uptake as "the duration for the deprotonated rotation" (indicated in green in **Figure 4a**). Again, several Monte Carlo steps may be required to achieve this proton uptake. We define the time from the arrival at the proton uptake angle to success of proton uptake as the "the duration for proton uptake" (blue in **Figure 4a**). Then, the $c$-subunit returns to the resting state. Thus, the entire time could be divided into three stages: stage 1, the duration for proton release; stage 2, the duration for deprotonated rotation; and stage 3, the duration for proton uptake, in addition to the resting time. Note that these durations are defined for each $c$-subunit and that the durations in one $c$-subunit

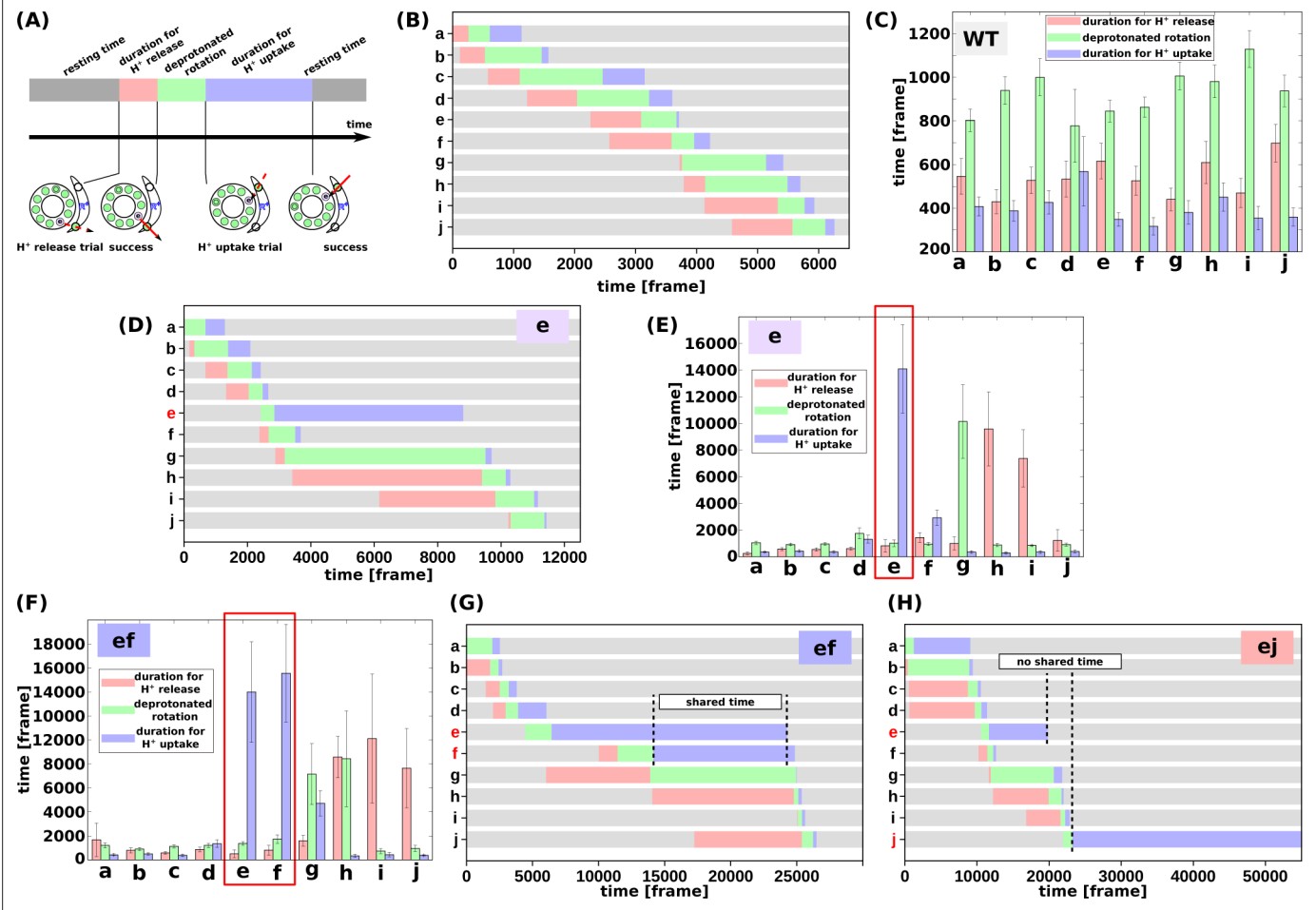

**Figure 4.** Analysis of the molecular simulations. (**a**) Schematic graph of duration times. Total time was divided into the duration for proton release, the duration for deprotonated rotation, the duration for proton uptake, and the resting time. (**b**) Representative time course of durations for the WT. (**c**) Histogram of durations for every *c*-subunit of the WT. (**d**) Representative time course of durations for the single mutant "e." (**e**) Histogram of durations for the single mutant "e." (**f**) Representative time course of durations for the double-mutant "ef." (**g**) Histogram of durations for the double-mutant "ef." (**h**) Representative time course of durations for the double-mutant "ej." WT, wild-type.

The online version of this article includes the following source data for figure 4:

**Source data 1.** Analysis of the molecular simulations.

overlap with durations in other *c*-subunits. For each mutant and for the WT, for each of the 10 *c*-subunits, we analyzed these three durations.

First, we examined the time course of a representative trajectory and the average durations for the WT for each *c*-subunit (***Figure 4b and c***). The average durations for stages 1, 2, and 3 were approximately 500, 1100, and 200 MD frames, respectively. As expected, there were no significant differences in the durations among the 10 *c*-subunits.

Next, we performed the same analysis using the single mutant "e" (***Figure 4d and e***). We found that the mutation in the *c*(e)-subunit clearly affects the duration for this subunit; stages 1 and 2 did not differ much from those in the WT, whereas the duration of stage three was much longer than that in the WT, since *c*E59D has a lower rate of proton transfer, and since the pKa value of *c*E59D is lower than that of *c*E59. The increased duration of stage two in the *c*(g)-subunit, and that of stage three in the *c*(h)- and *c*(i)-subunits, were caused by the delay in proton uptake of the *c*(e)-subunit. Because the *c*(e)-subunit scarcely received protons from the IMS channel, and because the *c*(e)-subunit often stopped around the IMS channel, the *c*(g)-subunit can rarely overcome the *a*-Arg barrier, thus increasing the duration of stage two in the *c*(g)-subunit. Similarly, the *c*(h)- and *c*(i)-subunits spend most of their time in the membrane and are unlikely to pass a proton to the *a*-subunit, but they approach the matrix

channel momentarily in fluctuations and try to release a proton to the *a*-subunit. However, the acceptance ratio is so small that the duration of stage three increased.

We then analyzed double mutants. For the "ef" mutant (*Figure 4f and g*), similarly to that for the "e" mutant, the *c*E59D mutation in the *c*(e)-subunit prolongs the duration for proton uptake. Additionally, mutation in the *c*(f)-subunit also prolongs the duration for proton uptake. Interestingly, as shown in *Figure 4f*, these prolonged durations in *c*(e)- and *c*(f)-subunits are shared. Thus, by overlapping the delayed steps, the overall slowdown in the "ef" double-mutant system is lower than that if the effects of the two mutations were independent. In other words, sharing the delayed times of multiple subunits reduces the overall delay. In comparison, we examined the durations for the "ej" double-mutant (*Figure 4h*).

As expected, mutations in the *c*(e)- and *c*(j)-subunits slow proton uptake in these subunits, although the durations are not shared. Therefore, we expect that there is no coupling between the *c*(e)- and *c*(j)-subunit mutations resulting in additive effects of the two mutations.

In summary, coarse-grained MD simulations qualitatively reproduced the effects of the single and double mutants found in biochemical assays and provided molecular interpretations of the coupling between two mutations. When the two mutations are in distant subunits of the *c*-ring, the effects of the two mutations are additive. In contrast, two mutations in neighboring subunits can result in overlapping of delays by the two mutations, leading to reduced effects of the two mutations.

## Discussion

In this study, we determined whether *c*-subunits function in a cooperative manner for the rotation of the $F_oF_1$ $c_{10}$-ring and assessed the mechanistic role of *c*-Glu (*c*E56) in this cooperation. We have demonstrated that the degree of cooperation between two *c*-subunits depends on the distance between the *c*E56D hetero-mutations at the proton-carrying site. The activity of $F_oF_1$ was significantly decreased, but not completely abolished, by a single *c*E56D mutation. This activity was further decreased by the second *c*E56D mutation; moreover, the activity was high when the two mutations were introduced into nearby *c*-subunits, and the activity decreased as the distance between the two mutations increased. To the best of our knowledge, this is the first study providing unambiguous evidence for the coupling between two *c*-subunits. Molecular simulations reproduced the major features of biochemical experiments on single and double mutants and further revealed the molecular mechanisms of the coupling. Sharing of the prolonged durations by mutations in neighboring *c*-subunits leads to coupling.

When the *c*E56D substitution was introduced in one of the *c*-subunits, ATP synthesis activity decreased substantially. In *E. coli* $F_oF_1$, ATP-driven proton pump activity was reported to be decreased after substitution of the conserved *c*Asp61 residue with Glu (*Miller et al., 1990*). Here, after *c*E56D substitution, we detected partial retention of not only proton pump activity but also of ATP synthesis activity. In contrast, *c*E56Q substitution in one of the *c*-subunits was found to eliminate ATP synthesis activity, ATP-driven proton pump activity, and DCCD-sensitive ATP hydrolysis activity (*Mitome et al., 2004*). In this study, ATP synthesis activity and ATP-driven proton pump activity were not completely lost when the carboxyl group of Glu was replaced with that of Asp. A comparison of this result with the *c*E56Q substitution results suggested that the presence of a carboxyl group capable of undergoing protonation and deprotonation is critical for rotation in the ATP synthesis direction coupled with proton transfer, and for the proton-transfer-coupled rotation induced by ATP hydrolysis. As changing the Glu side chain to an Asp side chain decreased activity, we concluded that subtle structural differences in the proton-binding site caused by the one-methylene-group difference in the sidechain length, together with the change in pKa, slowed the elementary process required for driving rotation.

In the $F_oF_1$s, carrying the *c*E56D mutation in two *c*-subunits, ATP synthase activity was high when the two introduced Asp residues were close to each other, and the activity decreased as the distance between the two mutations increased. If the kinetic bottleneck in the $c_{10}$-ring rotation was only in one step of one *c*-subunit, the same activity would appear among double mutations with different relative separations. Alternatively, even if the *c*-subunit plays multiple roles, if each role works independently, the same activity would be obtained, irrespective of the mutational position. However, the experimental results showed that the activity was decreased when the two mutations were introduced farther apart. Thus, the data unambiguously indicate that the kinetic bottleneck in the *c*-ring rotation contains multiple *c*-subunits.

According to previously proposed models, proton release at $c$-Glu, electrostatic interaction between $a$-Arg and $c$-Glu, and proton binding at $c$-Glu drive $c$-ring rotation (*Vik and Antonio, 1994*; *Elston et al., 1998*). Moreover, based on the crystal structure of mitochondrial $F_oF_1$, the $c$-subunits that face the $a$-Glu223 residue bridging a proton from IMS, the $a$-Arg residue involved in electrostatic interaction, and the $a$-Glu162 residue bridging a proton to the matrix are located apart; therefore, we hypothesized that the $c$-subunits on the $a$-Glu223 side of $a$-Arg play a role in proton release, whereas the $c$-subunits on the $a$-Glu162 side of $a$-Arg play a role in proton uptake in the ATP synthesis rotation. MC/MD simulations based on the $F_oF_1$ atomic structure have revealed that proton transfer causes $c_{10}$-ring rotation (*Kubo et al., 2020*). Here, MD simulation of $c_{10}$-ring rotation during ATP synthesis was performed based on the aforementioned hypothesis, that proton release and proton uptake are both affected by $c$E56D mutation. Our results indicated that the rotation speed is higher when the mutation is introduced at adjacent positions, and that the rotation speed decreases as the distance between the two mutants increases. These results, which are consistent with the findings of our biochemical experiments, indicate cooperative proton uptake during the rotation of the $c_{10}$-ring. Further analysis revealed that the waiting times for proton uptake in multiple subunits are shared. However, as the distance between the mutations increases, the degree of sharing of waiting time decreases, resulting in lower rotation speeds.

Overall, these findings suggest that at least three of the $c$-subunits on the $a/c$ interface cooperate during $c_{10}$-ring rotation in $F_o$. This is consistent with the presence of two or three deprotonated carboxyl residues facing the $a$-subunit in the MC/MD simulation of WT $F_oF_1$ (*Kubo et al., 2020*). In the WT, the $c$-subunit with deprotonated $c$E56 is considered to be the $c$-subunit waiting for proton uptake during ATP synthesis. Since the WT prefers pathways with two or three $c$-subunits waiting to uptake protons rather than only one $c$-subunit, the waiting time for proton uptake can be shared between two or three $c$-subunits. With respect to double mutation activity, the "ef" and "eg" mutants tended to have higher activity than the "eh–ej" mutants. This is consistent with the upper limit of three deprotonated $c$-subunits obtained from the WT simulations. The waiting time for protonation can be shared among three of sequential $c$-subunits of c(e), c(f), and c(g), but if they are located more than four subunits apart, the waiting time cannot be shared. Therefore, the activity of mutants "eh," "ei," and "ej" was lower than that of mutants "ef" and "eg."

One limitation of this study is that we used the fusion mutation and the $c$E56D mutation. These mutations may affect not only our hypothesized driving force but also other activities. However, we consider our interpretations of the results to be valid based on the comparison with the results of the same mutation combination, and the results of MD simulations. Second, our MC/MD model includes only the $a$-subunit and $c_{10}$-ring, whereas naturally occurring $F_oF_1$ also contains $F_1$ and the $b$-subunit. As $F_1$ exhibits threefold symmetry, which is mismatched with the tenfold symmetry in the $c_{10}$-ring, the entire $F_oF_1$ is expected to exhibit more complex and asymmetric behaviors, which can represent a direction for future investigation of the enzyme.

## Materials and methods
### Preparation of $F_oF_1$s carrying hetero mutations using fused multimeric $F_o$-$c$

Plasmids for the $F_oF_1$ mutants were generated from pTR19-ASDS (*Suzuki et al., 2002*) using the megaprimer method, and were then used for the transformation of a $F_o$-deficient *E. coli* strain, JJ001 (*Jones and Fillingame, 1998*). A plasmid for expressing the $F_oF_1$ mutant harboring a substitution of $F_o$-$c$ Glu-56 with Asp ($c$E56D) was prepared from pTR19-ASDS (*Suzuki et al., 2002*) using the megaprimer method; this yielded pTR19-CE56D. The $c$E56D mutation sequence was verified through DNA sequencing. $F_oF_1$ carrying a hetero-mutation of $c$E56D in a fused $c_{10}$-subunit prepared using Gly-Ser-Ala-Gly linkers (*Mitome et al., 2004*) was generated as follows. Briefly, an *Avr*II restriction site was introduced immediately after the initial $c$-subunit codon in the pTR19-CE56D expression plasmid, and new *Nhe*I and *Spe*I sites were introduced at downstream sites in the $F_o$-$c$ gene (to obtain pTR19-ACE56DN); pTR19-ACE56DN was digested with *Eco*RI and *Nhe*I, and the 1.3 kb *Eco*RI-*Nhe*I fragment was ligated into an *Eco*RI-*Avr*II site in pTR19-AC1N or pTR19-ACE56DN (to obtain pTR19-AC2DE or pTR19-AC2DD). Next, pTR19-AC2DE was digested with *Eco*RI and *Nhe*I, and the *Eco*RI-*Nhe*I fragment was ligated into an *Eco*RI-*Avr*II site in pTR19-AC1N or pTR19-ACE56DN

(to obtain pTR19-AC3DEE or pTR19-AC3DED). By using this procedure, *uncE* genes were singly fused to generate plasmids expressing six $F_oF_1$s containing tandemly fused decamers carrying the *c*E56D mutation at the first hairpin (mutant "e"), first and second hairpins (ef), first and third hairpins (eg), first and fourth hairpins (eh), first and fifth hairpins (ei), and first and sixth hairpins (ej). The multimer *uncE* genes of the mutants were verified through plasmid restriction mapping. Plasmids generated for the WT and mutant $F_oF_1$s were singly expressed in $F_o$-deficient *E. coli* strain JJ001 (*pyrE41, entA403, argHI, rspsL109, supE44, uncBEFH, recA56,* and *srl::Tn10*) (*Jones and Fillingame, 1998*). Transformants were cultured, and membrane vesicles were prepared as previously described (*Mitome et al., 2004*).

## Analytical procedures

ATPase activity was measured using an ATP-regenerating system at 37°C in 50 mM Hepes-KOH buffer (pH 7.5), containing 100 mM KCl, 5 mM $MgCl_2$, 1 mM ATP, 1 µg/ml FCCP, 2.5 mM KCN, 2.5 mM phosphoenolpyruvate, 100 µg/ml pyruvate kinase, 100 µg/ml lactate dehydrogenase, and 0.2 mM NADH (*Mitome et al., 2004*). One unit of activity was defined as hydrolysis of 1 µmol of ATP per minute; the slopes of decreasing 340 nm absorbance in the steady-state phase (400–600 s) were used for calculating activity. The sensitivity of ATP hydrolysis activity to DCCD-induced inactivation was analyzed as previously reported (*Noji et al., 2017*). The ATP hydrolysis activity in the presence of 0.1% lauryldimethylamine oxide was measured to estimate the amount of $F_oF_1$ in the membrane vesicles. ATP-driven proton pump activity was measured as the fluorescence quenching of ACMA (excitation/emission: 410/480 nm) at 37°C in 10 mM Hepes-KOH (pH 7.5), 100 mM KCl, and 5 mM $MgCl_2$, supplemented with membrane vesicles (0.5 mg protein/ml) and ACMA (0.3 µg/ml) (*Mitome et al., 2004*). The reaction was initiated by adding 1 mM ATP, and quenching reached a steady level after 1 min; after 5 min, FCCP (1 µg/ml) was added, and fluorescence reversal was confirmed. The magnitude of fluorescence quenching at 3 min relative to the level after FCCP addition was recorded as the proton pump activity. ATP synthesis activity was measured at 37°C using luciferase assays as previously described (*Mitome et al., 2017; Suzuki et al., 2007*). After incubating for 5 min at 37°C, we poured 1.6 ml PA3 buffer (10 mM Hepes-KOH [pH 7.5], 10% glycerol, and 5 mM $MgCl_2$), 2.5 mM KPi (pH 7.5), 0.53 mM ADP (Calbiochem, San Diego, CA), 26.6 µM $P^1,P^5$-di(adenosine-5′) pentaphosphate (Sigma-Aldrich, St. Louis, MO), 50 µg/ml inverted membranes, and 0.125 volumes CLS II solution (ATP Bioluminescence Assay Kit CLS II; Sigma-Aldrich) into the cuvettes; 0.5 mM NADH was added after starting the measurement. Synthesized ATP amounts were calibrated using a defined amount of ATP at the end of the measurement. FCCP addition was confirmed to prevent ATP synthesis. Specific activity was calculated as follows.

Using the data of the time course of the luminescence of luciferin-luciferase, the slope for 90 s before the addition of NADH was subtracted from the slope for 50 s after the addition of NADH. Defined amount of ATP was repeatedly added four times at 20-s intervals, and the average increase in luminescence per ATP amount over those times was calculated as the difference between the average luminescence for 5 s immediately after ATP addition and the average luminescence for 5 s immediately before ATP addition. The standard deviation and standard error were calculated.

ATP synthesis activity was calculated as follows: the slope of ATP synthesis deducted at the baseline was divided by the increase in luminescence per amount of ATP and membrane protein (mg) added. Error propagation processing was performed to calculate the standard deviation.

To estimate the amount of ATP synthase in the inverted membrane vesicle, the specific ATPase activity of the inverted membrane vesicles in 0.1% LDAO was calculated using the ATP regeneration system from the slope at 240–300 s after the addition of LDAO. Mean values and standard deviations were calculated from three or four data sets of each lot of the inverted membrane vesicles of mutants.

Since the expression level of ATP synthase differs depending on the mutant, the ATP synthesis activity of the inverted membrane assuming the same expression level of ATP synthase should be estimated. The "ei" and "ej" mutants showed relatively low ATP hydrolysis activity in the presence of LDAO corresponding to the expression level. The ATP synthesis activity was calculated assuming that the expression level corresponded to the average of the four mutants, "e," "ef," "eg," and "eh." In the actual calculation, the ATP synthesis activity of the inverted membrane of each mutant was divided by the ATP hydrolysis activity of the mutant in the presence of LDAO, and the average of the ATP hydrolysis activity of the "e," "ef," "eg," and "eh" mutants in the presence of LDAO was

multiplied. Error propagation processing was performed for calculating the standard deviation and standard error.

Mean values and standard deviations were calculated from nine datasets of three lots of mutants $c_{10}$, "e," and "eg," from 10 data sets of three lots of mutants "ef," "eh," and "ei," and from 8 data sets of three lots of mutant "ej." Error propagation processing was performed for standard deviation and standard error calculation. The unbiased estimate of variance was calculated from the standard deviation, and the Student's t-test was performed between the two mutants using the mean value and the unbiased estimate of variance to calculate the p-value.

Protein concentrations were determined using a BCA Assay Kit (Thermo Fisher Scientific, Waltham, MA), with bovine serum albumin serving as a standard. Membrane vesicles were separated using sodium dodecyl sulfate polyacrylamide gel electrophoresis (SDS-PAGE) with 15% gels containing 0.1% SDS, and proteins were stained with Coomassie Brilliant Blue R-250. $F_o F_1$ expression was confirmed by immunoblotting with anti-β and anti-c polyclonal antibodies for $F_o F_1$ from the thermophilic *Bacillus* PS3.

## Basic simulation system

To represent the proton transfer-coupled rotational motion of the $c_{10}$-ring, protein motion and proton jump were modeled using MD and MC, respectively, and these dynamics were combined to reproduce $c_{10}$-ring rotational motion with proton hopping (*Kubo et al., 2020*; *Figure 3—figure supplement 1a*). In our simulation system, we included the *a*-subunit and $c_{10}$-ring (*Figure 1a*) structure models of yeast $F_o$ based on the cryo-EM structure of a yeast mitochondrial ATP synthase (PDB ID: 6CP6) (*Srivastava et al., 2018*). The entire energy function was defined as

$$V_{total}(\boldsymbol{R}, \boldsymbol{H}^+) = V_{non-es}(\boldsymbol{R}) + V_{es}(\boldsymbol{R}, \boldsymbol{H}^+) + V_{pka}(\boldsymbol{H}^+)$$

where $R$ represents all the coordinates of the protein, $\boldsymbol{H}^+$ collectively represents the protonated state of 12 protonatable sites. The first term on the right-hand side, $V_{non-es}(R)$, is the whole-protein energy without the electrostatic interactions. We used the AICG2+ coarse-grained model to describe this energy function, where each amino acid is represented as a single particle located at the corresponding Cα atom. While the lipids were not explicitly modeled, the interactions between the protein residues and the lipid membranes were represented through an implicit membrane potential. Water solvents were also treated implicitly. The second term, $V_{es}(\boldsymbol{R}, \boldsymbol{H}^+)$, represents the electrostatic interaction that depends on both the protein coordinates and the protonated state. This is the sum of the regular Coulomb interaction, $V_C(\boldsymbol{R}, \boldsymbol{H}^+)$, between all electrostatic residues and a term, $V_{mem}(\boldsymbol{R}, \boldsymbol{H}^+)$, that depends on the membrane environment. Primarily, the term $V_{mem}(\boldsymbol{R}, \boldsymbol{H}^+)$ is a potential that applies for 10 cE59: When cE59 stays in the membrane region, the deprotonated state has a high energy. The validity of the membrane model, $V_{mem}(\boldsymbol{R}, \boldsymbol{H}^+)$, was discussed in the previous paper by *Kubo et al., 2020*. The last term, $V_{pka}(\boldsymbol{H}^+)$, expresses the energy required to protonate a protonatable residue. This energy depends on environmental conditions, such as membrane potential and pH, and the inherent pKa value. In this paper, we treated the membrane potential was to be $150 mV$, and the pH value of the IMS side and the matrix side to be 7.0 and 8.0, respectively. The hybrid MC/MD simulations consisted of the MC phase, at which protonation states of 12 protonatable sites (the glutamic acid [or aspartic acid in the case of mutants] in 10 c-subunits, aE223, and aE162) are updated, and the MD phase, when amino acid positions are updated by Langevin dynamics. Each round contained MC trial moves for all the protons involved, followed by $10^5$ MD steps (*Figure 3—figure supplement 1a*). All simulation setups were the same as those we have recently reported (*Kubo et al., 2020*), except for the treatment of the cE59D mutation.

## Treatment of cE59D in the simulation

In the hybrid MC/MD simulation, we mimicked cE59D mutations in the following manner. In the MD part, we simply changed the amino acid identity of the corresponding residue from glutamic acid to aspartic acid using the mutagenesis feature of PyMol (*Figure 3—figure supplement 1b*). Given the nature of our coarse-grained representation, this results in minor changes. The MC move represents proton transfer, which must be largely affected by the cE59D mutations via two distinct mechanisms, that is, the change in transfer efficiency and the change in the free energy difference between protonated and deprotonated states. For the former, the proton transfer efficiency is markedly reduced by the

cE59D mutation because aspartic acid has a shorter sidechain than glutamic acid by one methylene-group, and because the sidechain reorientation found in the corresponding glutamic acid (**Symersky et al., 2012**) may not occur in the aspartic acid mutant. In our model, the transfer efficiency contains $exp\left(-A\left(r-r_0\right)\right)$ factor, where $r$ is the distance between Cα atoms of the donor and the acceptor, the offset distance $r_0$ represents the sum of sidechain lengths of the donor and acceptor, and $A$ is the decay rate. We used $r_0 = 0.8nm$ for cE59 (the same value as reported previously **Kubo et al., 2020**) and set $r_0 = 0.6nm$ for cE59D, representing its shorter sidechain of aspartic acid. The decay rate $A$ was set to 2.5 (1/nm) for cE59 (the same value as reported previously **Kubo et al., 2020**) and 9.0 (1/nm) for cE56D, assuming the absence of sidechain reorientation in the mutant (**Figure 3—figure supplement 1c**). Second, the free energy difference between the states before and after the proton transfer is modulated by pKa differences in the donor and the acceptor amino acids and thus is affected by the cE59D mutation. Although the pKa value specific to the corresponding site is unknown, we empirically chose pKa=8.0 for cE59 and 7.0 for cE59D considering the intrinsic difference in pKa values, a previous argument (**Srivastava et al., 2018**), and computational estimates of pKa value by PROPKA (**Li et al., 2005**). The validity of the decay rate A and the pKa values are further discussed below.

## The validity of the simulation parameters

The decay rate A and pKa of cE59 were set to 2.5 (1/nm) and 8.0, respectively, which are the same values as those used by **Kubo et al., 2020**. In this study, for the mutant cE59D, we set the decay rate A, and the pKa of cE59D as 9.0 (1/nm) and 7.0, respectively. Here, we discuss the validity of these parameter selections for cE59D.

First, the parameter $A$ appears in the weight $w_{i\rightarrow j}\left(r,\theta\right)$ estimation that determines the proton hopping probability,

$$w_{i\rightarrow j}\left(r,\theta\right) = f\left(r\right)g\left(\theta\right)h_{R176}$$

in which $f\left(r\right) = exp\left(-A\left(r-r_0\right)\right)$ represents the distance dependence. $g\left(\theta\right)$ is introduced to reflect the sidechain orientation of cE59, and is a Gaussian function that depends on the angle, $\theta$, between the vector from the midpoint of the two adjacent residues of cE59 to cE59 itself, and of the vector from cE59 to the half-channel of a-subunit (aE223 and aE162 in $g\left(\theta\right)$ for the IMS and the matrix-side half-channels). The parameters in the Gaussian function are determined using the cryo-EM structure. $h_{R176}$ is included to mimic the role of aR176 that inhibits the proton leakage between the two half channels (**Mitome et al., 2010**).

Next, we demonstrated the effect of cE59D on proton transfer probability. Since the cE59D mutation does not affect the role of aR176, $h_{R176}$ should not be changed. The angle dependence g(θ) may be affected by the mutation. However, there is no direct structural information on the side chain of the mutated aspartic acid in its protonated/deprotonated states. Thus, we applied the same function as that for glutamic acid.

**Symersky et al., 2012** obtained the X-ray structure of the c-ring not embedded in the lipid bilayer and compared it with the previous c-ring structure embedded in the lipid bilayer (**Symersky et al., 2012**). They found that the side chain of cE59 in the lipid environment has its tip facing inside of the c-ring (closed conformation), whereas the side chain of cE59 in water aqueous solution has its tip facing outside of the c-ring (open conformation). In addition, MD simulations confirmed that the orientation of the side chain of cE59 is reversible depending on the environment. Based on these results, they concluded that the orientation of the side chains of cE59 changes when cE59 moves from the environment in the lipid membrane to the environment facing the a-subunit, which would facilitate proton hopping. It should be noted that the sidechain reorientation depends on the proximity of cE59 with its proton-relaying partner (aE223 or aE162 in our case). Thus, in our coarse-grained model, this effect can effectively be included in $f\left(r\right)$. For the case of the cE59D mutant, however, it is unlikely that the same degree of the reorientation occurs. Therefore, we decided to increase the decay rate A, of cE59D.

To examine the impact of this parametrization, we conducted the same simulation with the parameter A of cE59D set at 2.5/nm, the same value as in cE59. The results in **Figure 3—figure supplement 2a** show that the velocity of the single mutant (e) was lower than that of the WT, but the extent of decrease (30%) was smaller than that of the experimental result (75%). Therefore, we set A=9.0/nm as

**Table 3.** pKa values predicted by PROPKA.

| $c$-ring | a | b | c | d | e | f | g | h | i | j |
|---|---|---|---|---|---|---|---|---|---|---|
| $c$E59 | 7.40 | 6.10 | 5.89 | 5.89 | 5.89 | 5.89 | 5.89 | 6.73 | 8.03 | 7.34 |
| $c$E59D | 6.49 | 5.37 | 5.16 | 5.16 | 5.16 | 5.16 | 5.16 | 5.85 | 7.08 | 6.54 |

the parameter of $c$E59D, which results in consistent results being obtained in the experiment for the single mutant (e).

Next, we assessed the pKa values of $c$E59 and $c$E59D. The intrinsic pKa of aspartic acid in aqueous solutions is smaller than that of glutamic acid. Also, the pKa values of the donor and the acceptor inside proteins are thought to be larger in water due to pKa shift (*Srivastava et al., 2018*). Indeed, pKa estimations by a standard tool, PROPKA (*Li et al., 2005*), on the WT $c_{10}$-ring and the $a$-subunit complexes showed that the pKa values of the $a$-subunit-facing chain-a, -h, -i, and -j have particularly large pKa values, with the largest being 8.03 for chain-i (*Table 3*). Therefore, based on the study by (*Srivastava et al., 2018*) and the PROPKA result, we set the pKa of $c$E59 at 8.0. Similarly, we used PROPKA for assessing the pKa values in the complex of the $c_{10}$-ring with $c$E59D and the $a$-subunit, finding that the pKa value was, on average, 1.0 unit smaller than that in the $c$E59. Therefore, we set the pKa for $c$E59D at 7.0.

Since the difference in the pKa values between aspartic acid and glutamic acid in the aqueous solution was 0.2, we also considered an alternative parametrization; the pKa value of the $c$E59D at 7.8 (*Figure 3—figure supplement 2b*). The results showed no significant difference in the rotation velocity between the single mutant (e) and the double mutant (ej), which is inconsistent with the results of the experiment. Thus, we decided to consider the estimate made using PROPKA, and the pKa value for the $c$E59D was set at 7.0.

## Simulations and their analyses

For each of the WT $F_oac_{10}$ and the six $c$E59D mutation patterns corresponding to the biochemical assay, we carried out 10 independent simulation runs with different stochastic forces. The mutants included the single mutant "e" and the five double-mutants "ef," "eg," "eh," "ei," and "ej." The single mutant "e," for example, has the $c$E59D substitution only in the "e" chain, whereas other chains contain the WT $c$-subunit sequence. The double-mutant "ef" harbors substitutions in the two neighboring subunits. Each simulation run contained 6000 rounds of MC/MD cycles (twice as long as in our previous paper *Kubo et al., 2020*). Each round contained MC trial moves for all the protons involved and $10^5$ MD steps. Thus, the entire trajectory corresponds to $6.0 \times 10^8$ MD (60,000 frames saved).

Notably, due to limitations in the computation time, we could simulate only one to a few turns of 360° rotations for each trajectory. As the mutant systems show asymmetric arrangements, the unbiased estimate of average velocities requires the rotation of multiples of 360°. Thus, we used the cumulative rotation angle and the MD time step at which the $c_{10}$-ring returned to the initial orientation for the last time in each trajectory. The rotation velocity was obtained as the ratio of the cumulative rotation angle to the MD time. This velocity was then averaged over 10 trajectories.

## Acknowledgements

The authors thank Dr. Toshiharu Suzuki and Dr. Masasuke Yoshida for providing us with the expression plasmid for WT $F_oF_1$. This work was supported partly by a Grant-in-Aid for Scientific Research (C) and (B) [17K07922, 19H02577] and the Cooperative Research Program of "NJRC Mater. & Dev."

## Additional information

### Funding

| Funder | Grant reference number | Author |
|---|---|---|
| Japan Society for the Promotion of Science | 17K07922 | Noriyo Mitome |
| Japan Society for the Promotion of Science | 19H02577 | Noriyo Mitome |
| Network Joint Research Center for Materials and Devices | | Noriyo Mitome |

The funders had no role in study design, data collection and interpretation, or the decision to submit the work for publication.

### Author contributions

Noriyo Mitome, Conceptualization, Data curation, Formal analysis, Funding acquisition, Investigation, Methodology, Project administration, Resources, Supervision, Validation, Writing – original draft, Writing – review and editing; Shintaroh Kubo, Conceptualization, Data curation, Formal analysis, Investigation, Methodology, Software, Validation, Writing – original draft, Writing – review and editing; Sumie Ohta, Hikaru Takashima, Yuto Shigefuji, Data curation, Formal analysis, Investigation; Toru Niina, Data curation, Formal analysis, Investigation, Methodology, Software; Shoji Takada, Conceptualization, Data curation, Formal analysis, Investigation, Methodology, Project administration, Software, Supervision, Validation, Writing – original draft, Writing – review and editing

### Author ORCIDs

Noriyo Mitome (ID) http://orcid.org/0000-0001-5713-2598
Shintaroh Kubo (ID) http://orcid.org/0000-0002-0946-8879
Toru Niina (ID) http://orcid.org/0000-0003-3025-4418
Shoji Takada (ID) http://orcid.org/0000-0001-5385-7217

### Decision letter and Author response

Decision letter https://doi.org/10.7554/eLife.69096.sa1
Author response https://doi.org/10.7554/eLife.69096.sa2

## Additional files

### Supplementary files

• Transparent reporting form

### Data availability

All data generated or analysed during this study are included in the manuscript. Source data files have been provided for Figures 2, 3 and 4.

The following previously published datasets were used:

| Author(s) | Year | Dataset title | Dataset URL | Database and Identifier |
|---|---|---|---|---|
| Srivastava A.P, Luo M, Symersky J, Liao M.F, Mueller D.M | 2018 | Monomer yeast ATP synthase (F1Fo) reconstituted in nanodisc | https://www.rcsb.org/structure/6CP6 | RCSB Protein Data Bank, 10.2210/pdb6CP6/pdb |

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
