## [Editor Report]

F_o_F_1_-ATP synthase is a membrane enzyme that uses the proton motive force to synthesize ATP via rotation-coupled mechanism. The symmetry of this high-order homo-oligomeric enzyme suggests cooperativity among the protomers. Here, biochemical and molecular biology studies, and simulations demonstrate that energy transduction is indeed inherently cooperative in F_o_F_1_-ATP synthase.

---

## [Decision Letter]

**Decision letter after peer review:**

Thank you for submitting your article "Cooperation among *c*-subunits of F_o_F_1_-ATP synthase in rotation-coupled proton translocation" for consideration by *eLife*. Your article has been reviewed by 3 peer reviewers, including Nir Ben-Tal as the Reviewing Editor and Reviewer #1, and the evaluation has been overseen by Richard Aldrich as the Senior Editor. The following individual involved in review of your submission has agreed to reveal their identity: Hiroyuki Noji (Reviewer #2).

Essential revisions:

1. A main concerns is the accuracy of biochemical assays, ATP synthesis activity measurement and ATP-driven proton pumping activity measurement. To our knowledge, it is not easy to achieve highly accurate and precise biochemical assays such as error within a few % even if we use highly purified enzymes. In this paper, the authors reported very small experimental error: around 1 % or even less. However, we have not found description of the how the experimental errors were determined. The authors did not use purified enzymes but inverted vesicles of *E. coli* expressing the mutated enzymes. One of critical parameters for the accuracy is the quantification of the enzymes in the vesicles that were estimated from the decoupled ATP hydrolysis activity measurement. The error in this quantification should be substantially smaller than 1 % to achieve such a high accuracy. In addition, the ATP synthesis activity and ATP-driven proton pumping activity were measured from the time courses of the assays that should also include some experimental errors as found in the noise and drift in the time courses of proton pumping measurement (Figure 2c). Because the activity difference among the double mutants were subtle, the accuracy and precision of the biochemistry part are the critical points to prove the validity of their arguments. Detailed explanations on the estimation of experimental error as well as reproducibility are required.

2. Another concern is the validity of the simulation. The authors conduced Monte Carlo simulation for proton transfer step between c-subunit and a-subunit. The rate constant was represented in a simple exponential factor: exp(-A(r-r0)), where 'A' represents the decay rate, 'r' is physical distance between c-subunit and a-subunit and 'r0' is the offset value that represents the sum of sidechain length of the proton transferring residues on c-subunit and a-subunit. They assumed smaller 'r0' and larger 'A' for cE59D mutant. Although the smaller 'r0' would be reasonable considering the shorter side change of aspartic acid, the reason for higher 'A' for the mutant is not clear. In addition, different values for pKa were given to the glutamic acid in the wild type c subunit (cE59) and to the aspartic acid in the mutant (cE59D), without rationalization. These parameters should be critical for the simulation results. The validity of the different 'A' and pKa in the mutant should be explained.

3. Another concern is that cooperativity is shown here in case of mutants, but it is not so obvious how it relates to the WT enzyme. One clue, to which authors only briefly relate, is that according to their earlier simulations in WT the preferred pathway is when 2 or 3 Glu are unprotonated at any time rather than just one Glu being protonated/unprotonated. This kind of "cooperativity" in WT enzyme and its relation to the data presented should be discussed in greater detail.

Also, parts of the text, such as the Introduction, are not very clearly written and can be improved.

*Reviewer #1 (Recommendations for the authors):*

1. Show computations (Figure 3b) and experimental data (Figure 2d?) side by side to make it easy to evaluate the correspondence.

2. Simulation procedure should be explained in greater detail. Not just as a pointer to the previous publication.

3. "Although the pKa value specific to the corresponding site is unknown, we empirically chose pKa = 8.0 for cE59 and 7.0 for cE59D considering the intrinsic difference in pKa values.": These values are much higher than the pKa of these amino acids in solution (around 4). Maybe justify the values based on calculation of the pKa shifts? PROPKA or something.

*Reviewer #3 (Recommendations for the authors):*

l. 72 – there are now true atomic structures of F-ATPase, which should be quoted here (e.g. yeast, bovine, ovine).

Figure 1 – which structure was used for depiction? Quote PDB and species.

There is no comment at all why Glu to Asp mutation was used. Previous study from this group (ref 8) indicated that replacement of a single E with Q in c10 ring completely abolishes the activity, as discussed at the end of this manuscript. This should be mentioned at least briefly at a point in text when E to D mutations are introduced.

l. 170-175 – for mutants eh-ej there is no evidence for fluorescence quenching after ATP addition in Figure 2c, so there is no evidence for proton pumping activity. It is not clear at all why authors used quenching after FCCP addition as an apparent measure of pumping. In mutants ef-ej this signal is reversed compared to c10 and e. By the same token one could argue that eg-ej are more active than ef, as they show larger quenching after FCCP addition. It seems fair to say only that e to eg show decreased pumping as they show slow quenching after ATP addition, while eh-ej do not show any pumping.

Similarly, it is not clear if in Figure 2b eh-ej show genuine ATP synthesis or this is more of unspecific reaction. Authors could show here a control with their previous E56Q mutant which does not show such activity.

Figure 2 A – very poor blot, not clear at all that we are looking at a specific signal.

C – for c10 the steady level of fluorescence is not achieved, so it is incorrect to use it for comparison with other mutants.

Table 1 – would be good to include data for WT for comparison.

l. 249 – comment should be added on the increased values for g, h, i in Figure 4e, which are presumably linked to e behavior.

[Editors' note: further revisions were suggested prior to acceptance, as described below.]

Thank you for resubmitting your work entitled "Cooperation among *c*-subunits of F_o_F_1_-ATP synthase in rotation-coupled proton translocation" for further consideration by *eLife*. Your revised article has been reviewed by 2 peer reviewers and the evaluation has been overseen by Richard Aldrich as the Senior Editor, and a Reviewing Editor.

The manuscript has been improved but one remaining issue still needs to be addressed.

The core data of this manuscript is the ATP synthesis activities of the mutants with double cE56E mutations: ef, eg, eh, ei, and ej in Figure 2b, on which all of arguments and the simulations are based. The main concern was the precision and accuracy of this measurement. This is because the ATP synthesis activity of the double mutants are remarkably smaller than that of the reference sample, c10. In addition, the difference in the activity among the mutants are also quite small, in the range from 1-4 % when standardized with the activity of the reference sample. Therefore, the biochemical measurement should achieve extremely high precision and accuracy to quantify the differences in the activity among the double mutants.

In the original manuscript, the error bars were represented with standard deviations (s. d.) that ranged 3-10 % as coefficient of variance (c. v.), that were extraordinarily small as a biochemical assay. Furthermore, the ATP synthesis activity measurement has multiple steps for calibrations and data corrections: baseline correction, calibrations for ATP quantification, and quantifications of ATP synthase protein in the inversed vesicles. Each step should have some experimental errors. Therefore, this reviewer asked the authors to provide experimental data to verify such small error bars.

Along this request, the authors re-calculated standard deviations in the revised version, considering the experimental errors in quantifications of synthesized ATP molecules and the enzyme molecules in the inversed vesicles. Resultant errors are 17-30 % as c. v. for each mutant. Although lot-to-lot variance of inverted vesicles and errors in baseline correction are still not considered, the error estimation seems to be relatively valid.

Then, the arising question is "Are the differences among the double mutants statistically significant?". The revised manuscript does not mention this point at all, even though this is the most critical point and the basis of the main concern. The authors provide the p-values between double mutants that range from 10-68%, in the 'source data' file. At least, the authors should clearly mention the statistical uncertainties of the ATP synthesis activities of the mutants, providing the standard deviations (not standard errors) and p-values in the main text. Otherwise, readers are not able to judge the significance of differences in ATP synthesis activity among the mutants.

Arguments based on such subtle differences with large p-values might be misleading. The authors are asked to reconsider the statistical validity and to re-build the arguments as well as the title and abstract to avoid possible overstatements.

Lastly, we want to mention a point about data correction on ATP synthesis activity measurement from another angle: The fraction of coupled enzyme that can be estimated from DCCD sensitivity (Table. 1). When we consider the DCCD sensitivity in the quantification of 'coupled ATP synthase enzyme in the inverted vesicles', the resultant ATP synthesis activity of the mutants may differ from the present one, possibly showing a different trend. Did the authors correct the quantification of ATP synthase enzyme in the reversed vesicles with the DCCD sensitivity?

---

## [Author Response]

Essential revisions:1. A main concerns is the accuracy of biochemical assays, ATP synthesis activity measurement and ATP-driven proton pumping activity measurement. To our knowledge, it is not easy to achieve highly accurate and precise biochemical assays such as error within a few % even if we use highly purified enzymes. In this paper, the authors reported very small experimental error: around 1 % or even less. However, we have not found description of the how the experimental errors were determined. The authors did not use purified enzymes but inverted vesicles of *E. coli* expressing the mutated enzymes. One of critical parameters for the accuracy is the quantification of the enzymes in the vesicles that were estimated from the decoupled ATP hydrolysis activity measurement. The error in this quantification should be substantially smaller than 1 % to achieve such a high accuracy. In addition, the ATP synthesis activity and ATP-driven proton pumping activity were measured from the time courses of the assays that should also include some experimental errors as found in the noise and drift in the time courses of proton pumping measurement (Figure 2c). Because the activity difference among the double mutants were subtle, the accuracy and precision of the biochemistry part are the critical points to prove the validity of their arguments. Detailed explanations on the estimation of experimental error as well as reproducibility are required.

We thank the reviewers for this comment. In the original manuscript, we did not report on error propagation. In the revised manuscript, we have explained the estimation of experimental error and reproducibility. We have also included the source data for the calculations. (Figure 2b source data 1).

The calculation of ATP synthase activity was carried out as follows.

“Using the data of the time course of the luminescence of luciferin-luciferase, the slope for 90 seconds before the addition of NADH was subtracted from the slope for 50 seconds after the addition of NADH. […] The unbiased estimate of variance was calculated from the standard deviation, and the student’s t-test was performed between the two mutants using the mean value and the unbiased estimate of variance to calculate the p-value.”

We have amended the sentences as follows.

L120-122

“Importantly, across all five double mutants, the activity tended to decrease further as the distance between the two mutation sites increased.”

L149-152

“Moreover, among the five double mutants, “ef” showed the highest activity (19.7% of that of *c*_10_ F_o_F_1_), and the activity tended to decrease as the distance between the two mutations increased (“eg”: 18.3%; “eh”: 15.6%; “ei”: 15.0%; “ej”: 14.0%).”

2. Another concern is the validity of the simulation. The authors conduced Monte Carlo simulation for proton transfer step between c-subunit and a-subunit. The rate constant was represented in a simple exponential factor: exp(-A(r-r0)), where 'A' represents the decay rate, 'r' is physical distance between c-subunit and a-subunit and 'r0' is the offset value that represents the sum of sidechain length of the proton transferring residues on c-subunit and a-subunit. They assumed smaller 'r0' and larger 'A' for cE59D mutant. Although the smaller 'r0' would be reasonable considering the shorter side change of aspartic acid, the reason for higher 'A' for the mutant is not clear. In addition, different values for pKa were given to the glutamic acid in the wild type c subunit (cE59) and to the aspartic acid in the mutant (cE59D), without rationalization. These parameters should be critical for the simulation results. The validity of the different 'A' and pKa in the mutant should be explained.

We thank the reviewers for this comment. Indeed, the original manuscript did not explain, in depth, the reasoning behind the parameter choices for the cE59D mutant.

The decay rate, A, that appears in the proton transfer probability calculation represents the donor-acceptor distance dependence of the proton transfer. In the WT case, a previous structure study by Symersky et al. (Symersky et al. Nat Struct Mol. Biol. (2012)) suggests that the sidechain of the glutamic acid changes its orientation upon interaction with the proton-relaying residues in the *a*-subunit. Since this reorientation depends on the donor-acceptor distance, the sidechain reorientation can effectively be included in the evaluation of parameter *A*. On the other hand, the mutant cE59D is unlikely to show the same amount of the sidechain reorientation, which can be mimicked by the increased value of A, effectively; though no direct evidence of this is available. In our study, we chose a larger value of *A* when comparing velocity reduction in the single mutant ("e"). In this revision, to assess the impact of this parametrization, we performed extra-simulations with the decay rate A, for cE59D kept unchanged to that of cE59. The results in Figure 3—figure supplement 2 show that the results thus obtained were not consistent with those of the single-mutant ("e").

The pKa values of the proton-carrying glutamic acids and cE59D were determined based on their intrinsic values, as in a previous study by Srivastava et al., and estimated using a standard tool, PROPKA. In general, the pKa values of the proton donor and acceptor inside proteins are thought to be much larger than the intrinsic pKa values of their amino acids in aqueous solutions. Indeed, PROPKA estimated that the pKa value of cE59 that faces the *a*-subunit is ~8, and that the difference in pKa values between cE59 and cE59D is ~1 (Table 2). To assess the importance of the pKa values, in this revision, we performed extra-simulations with the pKa value of cE59D at 7.8, which is 0.2 smaller than that of cE59. We obtained inconsistent results with those of the biochemical assays (Figure 3—figure supplement 2).

In the revised manuscript, we have added a subsection describing the validity of the simulation parameters (subsection heading: The validity of the simulation parameters").

"For the former, the proton transfer efficiency is markedly reduced by the *c*E59D mutation because aspartic acid has a shorter sidechain than glutamic acid by one methylene-group and because the sidechain reorientation found in the corresponding glutamic acid^13^ (Symersky et al. Mol. Biol. (2012)) may not occur in the aspartic acid mutant."

"The decay rate was set to 2.5 (1/nm) for *c*E59 (the same value as reported previously^23^) and 9.0 (1/nm) for *c*E56D, assuming the absence of sidechain reorientation in the mutant (Figure 3—figure supplement 1c)."

"we empirically chose pKa = 8.0 for *c*E59 and 7.0 for *c*E59D considering the intrinsic difference in pKa values, a previous argument, and computational estimates of pKa value by PROPKA."

3. Another concern is that cooperativity is shown here in case of mutants, but it is not so obvious how it relates to the WT enzyme. One clue, to which authors only briefly relate, is that according to their earlier simulations in WT the preferred pathway is when 2 or 3 Glu are unprotonated at any time rather than just one Glu being protonated/unprotonated. This kind of "cooperativity" in WT enzyme and its relation to the data presented should be discussed in greater detail.Also, parts of the text, such as the Introduction, are not very clearly written and can be improved.

We appreciate reviewers for this comment. We have added the following explanation:

“In the WT, the *c*-subunit with deprotonated *c*E56 is considered to be the *c*-subunit waiting for proton uptake during ATP synthesis. Since the WT prefers pathways with two or three c-subunits waiting to uptake protons rather than only one *c*-subunit, the waiting time for proton uptake can be shared between two or three *c*-subunits. With respect to double mutation activity, the “ef” and “eg” mutants tended to have higher activity than the “eh-ej” mutants. This is consistent with the upper limit of three deprotonated *c*-subunits obtained from the WT simulations. The waiting time for protonation can be shared among three of sequential *c*-subunits of c(e), c(f), and c(g), but if they are located more than four subunits apart, the waiting time cannot be shared. Therefore, the activity of mutants “eh,” “ei,” and “ej” was lower than that of mutants “ef” and “eg”.”

In the introduction section in revised manuscript, we have added the following explanation:

“A more recent theoretical study using a hybrid Monte Carlo/molecular dynamics (MC/MD) simulation based on a high-resolution structure showed that there can be two or three deprotonated *c*-Glu residues facing the *a*-subunit concurrently^23^, suggesting that the waiting time for protonation of the deprotonated *c*-subunit is shared among the two or three of the *c*-subunits. However, the relationship between the sharing of deprotonation times among multiple *c*-subunits and the cooperativity of *c*-subunits for proton transport has not been discussed so far.”

Reviewer #1 (Recommendations for the authors):1. Show computations (Figure 3b) and experimental data (Figure 2d?) side by side to make it easy to evaluate the correspondence.

Following this advice, we have showed experimental data which is identical to figure 2b as figure 3c.

2. Simulation procedure should be explained in greater detail. Not just as a pointer to the previous publication.

Following this advice, we have markedly expanded the description of the simulation procedure, especially focusing on the proton-transfer processes.

In the revised manuscript, the following part has been added, together with a representation of the MD/MC procedure (Figure 3—figure supplement 1):

“The entire energy function was defined aswhere represents all the coordinates of the protein, collectively represents the protonated state of 12 protonatable sites. […] In this paper, we treated the membrane potential was to be , and the pH value of the IMS side and the matrix side to be and , respectively.”

3. "Although the pKa value specific to the corresponding site is unknown, we empirically chose pKa = 8.0 for cE59 and 7.0 for cE59D considering the intrinsic difference in pKa values.": These values are much higher than the pKa of these amino acids in solution (around 4). Maybe justify the values based on calculation of the pKa shifts? PROPKA or something.

We thank the reviewers for this comment. Following this advice, we have included the data from PROPKA.

Reviewer #3 (Recommendations for the authors):l. 170-175 – for mutants eh-ej there is no evidence for fluorescence quenching after ATP addition in Figure 2c, so there is no evidence for proton pumping activity. It is not clear at all why authors used quenching after FCCP addition as an apparent measure of pumping. In mutants ef-ej this signal is reversed compared to c10 and e. By the same token one could argue that eg-ej are more active than ef, as they show larger quenching after FCCP addition. It seems fair to say only that e to eg show decreased pumping as they show slow quenching after ATP addition, while eh-ej do not show any pumping.

We appreciate reviewer for this comment and for their advice.

In the original manuscript, since the degree of quenching after addition of ATP is small and the fluorescence shift after the addition of FCCP is relatively large, the fluorescence quenching degree was analyzed by assessing the fluorescence intensity immediately after the addition of FCCP and the fluorescence intensity at the steady level after the addition of FCCP. However, this analysis was not appropriate due to the inadequate time resolution of the experiment, and the unknown degree and time constant of fluorescence shift by the addition of FCCP. Figure2d has been deleted and the description has been revised, as follows:

Mutants “e,” “ef,” and “eg” showed proton pumping, indicated by the slow quenching after ATP addition, while mutants “eh,” “ei,” and “ej” did not show any pumping. The proton pump activity of the single mutant “e” was higher than that of the double mutants “ef” and “eg”.

Similarly, it is not clear if in Figure 2b eh-ej show genuine ATP synthesis or this is more of unspecific reaction. Authors could show here a control with their previous E56Q mutant which does not show such activity.

We thank the reviewer for this comment. We have added the mutant *c*_10_(E56Q) data to Figure 2b.

We have also added the explanation of the result of *c*_10_(E56Q) to the text of revised manuscript, and to the legend of Figure 2, as follows:

“For comparison, the *c*_10_(E56Q)-F_o_F_1_ with only a single E56Q mutation introduced into the first hairpin unit of the *c*_10_ did not catalyze ATP synthesis^8^.”

In the legend of figure 2,

“The rightmost bars [E56Q] show the results of *c*_10_(E56Q)-F_o_F_1_. Error bar: standard error.”

Figure 2 A – very poor blot, not clear at all that we are looking at a specific signal.C – for c10 the steady level of fluorescence is not achieved, so it is incorrect to use it for comparison with other mutants.

Thank you for your comment.

The following sentences have been added to describe Figure 2A.

“Unlike in the WT, there was no band of monomer *c*-subunits in the mutants, and relatively stronger bands were seen at the position of the *c*_10_ subunit, indicating that the *c*_10_ subunit of the mutants was expressed in the membrane (Figure 2a).”

As a response to secondary issue 1, we have deleted Figure 2d. Instead, we have added the following description to the text:

“Mutants “e,” “ef,” and “eg” showed proton pumping, indicated by the slow quenching after ATP addition, while mutants “eh,” “ei,” and “ej” did not show any pumping. The proton pump activity of the single mutant “e” was higher than that of the double mutants “ef” and “eg”.”

Table 1 – would be good to include data for WT for comparison.

Following this advice, we had added WT data to Table 1.

l. 249 – comment should be added on the increased values for g, h, i in Figure 4e, which are presumably linked to e behavior.

We appreciate reviewers for this comment. We have added the following description:

“Next, we performed the same analysis using the single mutant “e” (Figure 4d, e). […] However, the acceptance ratio is so small that the duration of stage 3 increased.”

[Editors' note: further revisions were suggested prior to acceptance, as described below.]

The manuscript has been improved but one remaining issue still needs to be addressed.The core data of this manuscript is the ATP synthesis activities of the mutants with double cE56E mutations: ef, eg, eh, ei, and ej in Figure 2b, on which all of arguments and the simulations are based. The main concern was the precision and accuracy of this measurement. This is because the ATP synthesis activity of the double mutants are remarkably smaller than that of the reference sample, c10. In addition, the difference in the activity among the mutants are also quite small, in the range from 1-4 % when standardized with the activity of the reference sample. Therefore, the biochemical measurement should achieve extremely high precision and accuracy to quantify the differences in the activity among the double mutants.In the original manuscript, the error bars were represented with standard deviations (s. d.) that ranged 3-10 % as coefficient of variance (c. v.), that were extraordinarily small as a biochemical assay. Furthermore, the ATP synthesis activity measurement has multiple steps for calibrations and data corrections: baseline correction, calibrations for ATP quantification, and quantifications of ATP synthase protein in the inversed vesicles. Each step should have some experimental errors. Therefore, this reviewer asked the authors to provide experimental data to verify such small error bars.Along this request, the authors re-calculated standard deviations in the revised version, considering the experimental errors in quantifications of synthesized ATP molecules and the enzyme molecules in the inversed vesicles. Resultant errors are 17-30 % as c. v. for each mutant. Although lot-to-lot variance of inverted vesicles and errors in baseline correction are still not considered, the error estimation seems to be relatively valid.

Thank you for evaluating our manuscript and for your constructive comments and suggestions. In the revised manuscript, we considered the lot-to-lot variance of inverted vesicles and baseline correction errors.

Regarding dispersion between the lots, ATP synthesis activity of the inverted membrane vesicles was analyzed for three lots, as shown in Figure 2b.

In the original version of the manuscript, the ATP synthesis activity of one lot, in which the amount of ATP synthease in the membrane was quantified using ATP hydrolysis activity with LDAO, was shown in Figure 2b.

The two other lots were not analyzed for ATPase activity with LDAO, so that these data were not shown. In the revised manuscript, the ATPase activities with LDAO of the other two lots were analyzed as well, and we calculated ATP synthesis activities using these three lots. We have revised in figure 2b, table 1 and lines 135-146 in the result section and lines 424-425, 437-439 in method section.

Then, the arising question is "Are the differences among the double mutants statistically significant?". The revised manuscript does not mention this point at all, even though this is the most critical point and the basis of the main concern. The authors provide the p-values between double mutants that range from 10-68%, in the 'source data' file. At least, the authors should clearly mention the statistical uncertainties of the ATP synthesis activities of the mutants, providing the standard deviations (not standard errors) and p-values in the main text. Otherwise, readers are not able to judge the significance of differences in ATP synthesis activity among the mutants.Arguments based on such subtle differences with large p-values might be misleading. The authors are asked to reconsider the statistical validity and to re-build the arguments as well as the title and abstract to avoid possible overstatements.

We have reconsidered the statistical validity of the ATP synthesis activities of the mutants and provided standard deviations and p-values in figure 2b, table 1 and lines 135-140 in the main text.

In the statistical processing between double mutants, a regression analysis was performed between the distance between mutations indicated by the number of *c*-subunits and the ATP synthase activity; the corresponding P value is shown in lines 140-146 in the text.

We have been deleted “Simple kinetic analysis of hetero-mutant experiments section” because the statistical significance of the experimental data could be insufficient for the analysis and discussion.

Lastly, we want to mention a point about data correction on ATP synthesis activity measurement from another angle: The fraction of coupled enzyme that can be estimated from DCCD sensitivity (Table. 1). When we consider the DCCD sensitivity in the quantification of 'coupled ATP synthase enzyme in the inverted vesicles', the resultant ATP synthesis activity of the mutants may differ from the present one, possibly showing a different trend. Did the authors correct the quantification of ATP synthase enzyme in the reversed vesicles with the DCCD sensitivity?

We did not correct ATP synthase in inverted membrane vesicles with the DCCD sensitivity.

For wild-type enzymes, DCCD sensitivity has been used as an indicator for coupled enzymes, but for enzymes with mutations in the DCCD binding site, such as cE56D, DCCD is inappropriate as an indicator for quantification.

The effect of DCCD sensitivity on cE56D remains unknown, and it is unclear whether DCCD binds to cE56D or whether cE56D affects DCCD binding of the adjacent cE56.

One possible explanation of the low sensitivity of cE56D to DCCD is that cE56D resists DCCD binding. In this case, the DCCD-insensitive fraction in ATPase activity may also exhibit ATP synthesis and proton pump activities, in which the quantification of coupled enzymes via DCCD sensitivity could be inappropriate.